# Combined Treatment with MEK and mTOR Inhibitors is Effective in In Vitro and In Vivo Models of Hepatocellular Carcinoma

**DOI:** 10.3390/cancers11070930

**Published:** 2019-07-03

**Authors:** Xianqiong Liu, Junjie Hu, Xinhua Song, Kirsten Utpatel, Yi Zhang, Pan Wang, Xinjun Lu, Jie Zhang, Meng Xu, Tao Su, Li Che, Jingxiao Wang, Matthias Evert, Diego F. Calvisi, Xin Chen

**Affiliations:** 1Pharmacy Faculty, Hubei University of Chinese Medicine Wuhan, Wuhan 430065, China; 2Department of Bioengineering and Therapeutic Sciences and Liver Center, University of California, San Francisco, CA 94143, USA; 3Institute of Pathology, University of Regensburg, Regensburg 93053, Germany; 4Key Laboratory of Biorheological Science and Technology, Ministry of Education, College of Bioengineering, Chongqing University, Chongqing 400044, China; 5Beijing Advanced Innovation Center for Food Nutrition and Human Health, College of Food Science and Nutritional Engineering, China Agricultural University, Beijing 100083, China; 6Department of Hepatic Surgery, the First Affiliated Hospital, Sun Yat-sen University, No. 58 Zhongshan 2nd Road, Guangzhou 510080, China; 7Department of Thoracic Oncology II, Key Laboratory of Carcinogenesis and Translational Research (Ministry of Education), Peking University Cancer Hospital & Institute, Beijing 100142, China; 8Department of General Surgery, the Second Hospital of Xi’an Jiaotong University, Xi’an Jiaotong University, Xi’an 710061, China

**Keywords:** hepatocellular carcinoma, MEK inhibitor, pan-mTOR inhibitor, PD901, MLN0128, cell proliferation, apoptosis

## Abstract

*Background*: Hepatocellular carcinoma (HCC) is the most common primary liver cancer histotype, characterized by high biological aggressiveness and scarce treatment options. Recently, we have established a clinically relevant murine HCC model by co-expressing activated forms of v-akt murine thymoma viral oncogene homolog (AKT) and oncogene c-mesenchymal-epithelial transition (c-Met) proto-oncogenes in the mouse liver via hydrodynamic tail vein injection (AKT/c-MET mice). Tumor cells from these mice demonstrated high activity of the AKT/ mammalian target of rapamycin (mTOR) and Ras/ Mitogen-activated protein kinase (MAPK) signaling cascades, two pathways frequently co-induced in human HCC. Methods: Here, we investigated the therapeutic efficacy of sorafenib, regorafenib, the MEK inhibitor PD901 as well as the pan-mTOR inhibitor MLN0128 in the AKT/c-Met preclinical HCC model. *Results*: In these mice, neither sorafenib nor regorafenib demonstrated any efficacy. In contrast, administration of PD901 inhibited cell cycle progression of HCC cells in vitro. Combined PD901 and MLN0128 administration resulted in a pronounced growth constraint of HCC cell lines. In vivo, treatment with PD901 or MLN0128 alone moderately slowed HCC growth in AKT/c-MET mice. Importantly, the simultaneous administration of the two drugs led to a stable disease with limited tumor progression in mice. Mechanistically, combined mitogen-activated extracellular signal-regulated kinase (MEK) and mTOR inhibition resulted in a stronger cell cycle inhibition and growth arrest both in vitro and in vivo. *Conclusions*: Our study indicates that combination of MEK and mTOR inhibitors might represent an effective therapeutic approach against human HCC.

## 1. Introduction

Hepatocellular carcinoma (HCC) is the most frequent form of primary liver cancer and the second leading cause of cancer-related mortality around the world [1,2,3]. The incidence of HCC has been rapidly and steadily rising over the past decades. Generally, patients with HCC do not show early stage symptoms and the diagnosis often comes late, thus most patients do not meet the criteria for effective surgical resection or liver transplantation [1,4]. For patients with advanced stage HCC, the multi-kinase inhibitor sorafenib is the standard of care worldwide. For those who progress on sorafenib, additional drugs, such as regorafenib, another multi-kinase inhibitor, have been recently approved as the second line therapeutics against HCC [5,6,7]. Recently, immune checkpoint inhibitors, including nivolumab and pembrolizumab, have demonstrated efficacy in ~20% of HCC patients [8]. However, many patients do not respond to any of these treatments and the overall survival rate of HCC remains extremely poor, with HCC incidence rate roughly coinciding with that of mortality [9]. Thus, novel and effective therapeutic strategies are of prime importance for this malignancy.

The phosphoinositide-3-kinase (PI3K)/AKT/mTOR pathway is active in diverse tumor entities, such as breast cancer [10], colon cancer [11], and cholangiocarcinoma [12]. Aberrant activation of this signaling cascade has also been found in about 40–50% of HCCs [13]. Evidence shows that this pathway plays an important role in cell proliferation, survival, and energy metabolism, and is associated with tumor lower differentiation, poorer prognosis, and rapid cancer recurrence [14,15]. Due to the key function of the PI3K/AKT/mTOR cascade in liver cancer, its suppression by targeted agents is a rational direction for the treatment of HCC. MLN0128 is a second-generation mTOR ATP site inhibitor [16] and can reduce the tumor burden effectively in CD44 expressing HCC, which is insensitive to sorafenib [17]. In addition, MLN0128 is currently under evaluation in several Phase I and II clinical trials, including a Phase I/II clinical trial as a first-line single agent compared with standard sorafenib in advanced HCC (NCT 02575339, https://clinicaltrials.gov/).

The MEK signaling is a critical molecular axis driving various cellular processes including growth, differentiation, survival, migration, and angiogenesis [18,19,20]. Either activating mutations of various oncogenes or growth factors are able to trigger this pathway [21]. Deregulation of the MEK signaling cascade has been described in several cancer types, including breast, melanoma, lung, and pancreatic tumors [19,20,21]. In light of this evidence, targeting this kinase offers an attractive therapeutic target for cancer and, consequently, various MEK inhibitors have been developed [21]. In human HCC, it has been shown that the Ras/MEK pathway is ubiquitously activated [22], and targeting MEK has shown to be detrimental for the growth of HCC cell lines [23]. Taken together, these data support the potential importance of MEK inhibition in HCC therapy.

We have recently established a clinically relevant murine HCC model by simultaneously overexpressing activated AKT and c-MET proto-oncogenes in the mouse liver (AKT/c-MET) by hydrodynamic tail vein injection. In these mice, pure HCC develop, and mice require to be sacrificed by 8 weeks post hydrodynamic injection due to high tumor burden. At the molecular level, AKT/c-MET tumor cells demonstrated high levels of activation of the AKT/mTOR and Ras/MAPK cascades [24]. 

In the present study, we investigated the therapeutic efficacy of sorafenib, regorafenib, the MEK inhibitor PD-0325901 (PD901), and the pan-mTOR inhibitor MLN0128 in vitro using HCC cell lines and in vivo using the AKT/c-MET preclinical HCC model. We found that a combination therapy targeting concurrently the Ras/MAPK and AKT/mTOR cascades is effective in inducing tumor growth restraint. Thus, our study underlines the synergistic efficacy of Ras/MAPK and AKT/mTOR inhibitors-based treatment in suppressing HCC growth, representing a new and promising therapeutic strategy for advanced HCC. 

## 2. Results

### 2.1. Limited Efficacy of Sorafenib and Regorafenib in HCC from AKT/c-MET Mice

Recently, we established a clinically relevant HCC model by hydrodynamically transfecting activated forms of AKT and c-MET proto-oncogenes (AKT/c-MET) in the mouse liver [24]. Specifically, we co-expressed myristoylated/activated (myr)-AKT and c-MET oncogenes together with the sleeping beauty transposase into the mouse liver via hydrodynamic tail vein injection. This procedure leads to the formation of high HCC burden within 8 weeks post-injection with 100% penetrance in AKT/c-MET injected mice [24]. This mouse HCC model has been replicated independently by other groups of scientists recently [25,26]. Since sorafenib remains the first-line treatment for HCC, and regorafenib is the second line drug for patients who progress with sorafenib, we tested these drugs in the AKT/c-MET murine HCC model (Figure 1A). 

Tumor growth was monitored in AKT/c-MET mice until 4 weeks after injection, when the mice display a moderate tumor burden (average liver weight ~4 g) (Figure 1A,B). Subsequently, AKT/c-MET mice were randomly separated into three cohorts. A group of mice at 4 weeks post-injection was harvested as a ‘pre-treatment’ cohort, while the remaining two groups were continually treated with either vehicle or sorafenib for 3 weeks (Figure 1A). Interestingly, we found that tumor continued to grow with sorafenib (30 mg/kg/day) treatment. All vehicle as well as sorafenib-treated mice had to be euthanized by 3 weeks of treatment due to high liver tumor burden. In AKT/c-MET mice, tumor nodules were diffused and colliding, with no surrounding capsules; as a consequence, it was impossible to accurately count the surface tumor nodule number in these mice (Figure 1C, right panels). As most (over 90%) of the liver parenchyma was occupied by the tumor cells, we used overall liver weight as the measure of tumor burden. This method has been shown to accurately reflect HCC burden in this murine liver tumor model by independent groups [25,26]. We found that the sorafenib- treated cohort had higher tumor burden than the pre-treatment cohort, and similar tumor burden was found in sorafenib- and vehicle-treated mice (Figure 1B,C). At the molecular level, sorafenib did not inhibit p-ERK or p-AKT expression in the mouse liver tissues (Figure 1D). At the cellular level, sorafenib treatment did not affect HCC cell proliferation, but was able to induce apoptosis (Figure 2A,B). However, as the cell apoptosis rate was relatively low even in sorafenib-treated mice, it was not able to significantly counteract the rapid tumor cell proliferation and, thus, had limited impact on overall tumor burden. Since sorafenib has been shown to inhibit VEGFR-mediated angiogenesis, we examined the microvasculature in sorafenib-treated HCC samples. Again, no significant differences were observed when comparing the vessel density in sorafenib- and vehicle-treated mice (Figure 2C).

Next, we treated the AKT/c-MET tumor bearing mice with regorafenib. Unexpectedly, we found that regorafenib was highly toxic to the mice, even at 15 mg/kg/day concentration. All mice treated with regorafenib showed signs of lethargy and profound weight loss. Due to these signs of overt toxicity, all regorafenib-treated mice had to be euthanized within ~1 weeks of treatment per the IACUC protocol (Xianqiong Liu and Xin Chen, University of California, San Francisco, CA, USA, Personal communication, 2018).

In summary, our study indicates that neither sorafenib nor regorafenib are effective against hepatocarcinogenesis induced by AKT/c-MET co-expression in mice, due to either lack of efficacy or significant toxicity. The lack of therapeutic potential exerted by sorafenib and regorafenib on tumor growth in AKT/c-MET mice is consistent with the clinical observation that these drugs have the limited efficacy in significant subsets of patients with advanced HCC.

### 2.2. Increased Growth Inhibition in Human HCC Cell Lines by PD901 and MLN0128 

As activated AKT/mTOR and Ras/MAPK signaling cascades are frequently and concomitantly observed in human HCC [24] as well as in AKT/c-MET hepatocellular lesions [24], we hypothesized that MEK and/or AKT/mTOR inhibitors might be effective for HCC treatment.

As a first step to test this hypothesis, we investigated the growth suppressive potential of the MEK inhibitor PD901 and the pan-mTOR inhibitor MLN0128 in human HCC cell lines. We found that the HCC cells tested were more sensitive to MLN0128, with IC50 ranging between 0.2 to 5 µM, when compared to PD901, which displayed a higher IC_50_, between 100 and 200 µM (Figure 3A,B). Importantly, when the HCC cell lines were treated with both PD901 and MLN0128 inhibitors, a significantly stronger growth suppressive activity was detected (Figure 3C).

At the molecular level, the levels of mTORC2 target phosphorylated/activated p-AKTS473, the mTORC1 target phosphorylated/activated p-RPS6 as well as phosphorylated/activated p-mTOR were strikingly reduced following MLN0128 administration in all HCC cell lines tested, whereas inconsistent results were detected when assessing the levels of phosphorylated PI3K (Figure 4). On the other hand, PD901 remarkably reduced the levels of phosphorylated/activated p-ERK (Figure 4). Deregulation of cell cycle results in unconstrained cell division, leading to continuous proliferation, and represents a pivotal driver of carcinogenesis [27]. We found that the expression of Cyclin D1, one of the critical proteins promoting cell cycle progression, was suppressed both in PD901 and MLN0128 treated HCC cells. Moreover, PD901 and MLN0128 combined treatment led to further decreased levels of Cyclin D1 in the HCC cells (Figure 4). No consistent changes of the cell cycle negative regulators, such as p53, p21, and p16, were observed in the same HCC cell lines (Figure 4).

We further investigated how these drugs affected HCC cell cycle progression. In all 3 HCC cell lines tested, PD901 induced cell cycle arrest, leading to the decreased cell numbers in S-phase, while MLN0128 had different effects depending on the cell line examined, with decreased cell numbers in S-phase in SNU475 and MHCC97H cells, but not Huh7 cells (Figure 5). Importantly, combined PD901 and MLN0128 treatment resulted in a more pronounced cell cycle arrest in all HCC cell lines tested when compared with single treatments (Figure 5).

Subsequently, we evaluated apoptosis in the three cell lines subjected to PD901, MLN1028, and combined treatment (Figure 6). We found that both PD901 and MLN1028 administration induced significant higher cell death than treatment with solvent (DMSO) alone at both time points examined. In all three cell lines, the apoptotic power of MLN0128 was significantly stronger than that of PD901. Of note, the combined administration of the two inhibitors did not result in a consistent significant increase of apoptosis when compared with treatment with single agents (Figure 6). SNU475 cells showed a marginal increased apoptosis in the combination treatment group both at 24 h and 48 h treatment. As concerns Huh7 cells, there was no significant increased apoptosis in the combination group at 24 h and 48 h time point. In MHCC97H cells, on the other hand, concomitant PD901 and MLN0128 administration led to a rise in apoptosis rate 48h after treatment (Figure 6).

Altogether, the present findings indicate that combined PD901/MLN0128 treatment induces a strong growth inhibition of HCC cells in vitro, predominantly by triggering cell cycle arrest.

### 2.3. PD901 and MLN0128 Combination Therapy Results in a Stable Disease in AKT/c-MET Mice

Our in vitro findings indicate that combined PD901/MLN0128 treatment leads to a strong growth suppression in human HCC cells. Subsequently, we investigated whether the same effects could be observed in vivo in the AKT/c-MET HCC preclinical model. Thus, AKT/c-MET tumor bearing mice were treated with PD901, either alone or in combination with MLN0128.

First, we evaluated the maximum dose of PD901 and MLN0128 that could be tolerated by mice. Our previous studies demonstrated that there is no significant toxicity dosing mice with 10mg/kg/day PD901 [28] or 1 mg/kg/day MLN0128 [29]. However, using mouse body weight as measurement of overall drug toxicity, dosing combined PD901 and MLN0128 at 10 mg/kg/day and 1 mg/kg/day separately to the mice for 5 days induced intolerable toxicity. Upon decreasing MLN0128 dose to 0.5 mg/kg/day, we found that 10 mg/kg PD901 plus 0.5 mg/kg MLN0128 was well-tolerated and, therefore, selected for the in vivo studies.

Similar to that described for the experiments with sorafenib (Figure 1A,B), tumor growth was monitored in AKT/c-MET mice until 4 weeks after injection (Figure 7A,B). Next, AKT/c-MET mice were randomly separated into five cohorts. A group of mice at 4 weeks post-injection was harvested as ‘pre-treatment’ cohort, while the remaining four groups were continually treated with vehicle, PD901, MLN0128, or PD901/MLN0128 for 3 weeks (Figure 7A). Total liver weight was used as the measurement of HCC tumor burden in mice. We found that MLN0128 or PD901 single treatment led to slower tumor growth in AKT/c-MET mice, as demonstrated by a significant lower tumor burden than the vehicle cohort (Figure 7B). The data also showed that the tumor burden of MLN0128 or PD901 monotherapy group was still higher than the pre-treatment group (Figure 7B), indicating the continual tumor growth despite of the therapy employed. In contrast, combined PD901 and MLN0128 administration exhibited a significantly improved therapeutic efficacy when compared with MLN0128 or PD901 monotherapy. Specifically, lowest liver weight was observed in PD901/MLN0128 combination therapy group (Figure 7B). Importantly, no difference in liver weight between the pre-treatment and combination therapy group was detected (Figure 7B). These results were also verified by macroscopic evaluation and histopathological analysis (hematoxylin and eosin staining) of the livers (Figure 7C). All tumor cells were positive for HA-tag, which stained the ectopically expressed AKT. Small tumor lesions were observed in pre-treatment as well as PD901/MLN0128 treated mice, whereas large lesions were found in vehicle, PD901, and MLN0128 treated mice (Figure 7C).

In summary, the present data indicate that, in the AKT/c-MET preclinical HCC model, PD901 and MLN0128 monotherapy led to progressive disease, although HCC grew at a slower rate, whereas combined PD901/MLN0128 treatment induced a stable disease.

### 2.4. Combined PD901/MLN0128 Regimen Inhibits Tumor Cell Proliferation In Vivo

Since we have demonstrated that PD901/MLN0128 combination treatment dramatically inhibits HCC cell proliferation in vitro (Figure 5), we asked whether the tumor stabilizing efficacy in vivo was also driven by this mechanism. Using Ki-67 immunohistochemistry as a surrogate marker of proliferation, we evaluated the proliferation indices in the five cohorts of AKT/c-MET mice. The data obtained from the analysis revealed that administration of PD901 or MLN0128 alone significantly decreased cell proliferation rates when compared with vehicle group (Figure 8A). Strikingly, PD901/MLN0128 combination treatment inhibited tumor cell proliferation more effectively than either PD901 or MLN0128 monotherapy (Figure 8A). As for the apoptosis rate, we discovered that MLN0128 monotherapy as well as combined PD901/MLN0128 treatment led to equivalent increase in apoptosis, whereas PD901 had no effect on tumor cell death (Figure 8B). However, the overall apoptosis rate was relatively low compared to cell proliferation rate. Therefore, the increased apoptosis is likely to have limited effects on overall tumor growth. Subsequently, to determine angiogenesis in the five mouse cohorts, we examined the expression of the CD34 protein (Figure 8C). Compared to the vehicle group, MLN0128 or PD901 monotherapy as well as combination treatment group led to a slight decrease of CD34 immunoreactivity, indicating that inhibition of angiogenesis is not a major mechanism for the anti-tumor activities exerted by MLN0128 or PD901 in AKT/c-MET mice. 

Mechanistically, we found that PD901 could profoundly decrease the expression of p-ERK1/2, the biomarker of PD901 efficacy, in PD901-treated mice, while MLN0128 treatment induced a decline in p-RPS6, p-4EBP1, p-mTOR and p-AKT expression (Figure 9). The expression of p-RPS6 was further inhibited after combined PD901/MLN0128 treatment (Figure 9). As concerns proliferation markers, PD901 administration reduced PCNA expression, whereas MLN0128 inhibited Cyclin D1 levels. Combined PD901/MLN0128 treatment led to decreased levels of both PCNA and Cyclin D1 (Figure 9). 

Overall, our study demonstrates that combined PD901/MLN0128 treatment strongly inhibits tumor cell proliferation, leading to stable disease in AKT/c-MET HCC mice.

## 3. Discussion

Progressed, unresectable HCC is a highly pernicious tumor with few systemic therapeutic options [1,4]. Multi-kinase inhibitors, such as sorafenib and regorafenib remain the first- and second-line regimens for patients with advanced HCC, respectively. However, the response to these drugs is very limited, leading to an increase of the overall survival only of a few months [9]. Indeed, in the clinical studies on sorafenib for advanced HCC, the overall radiological progression time was about 5.5 months in sorafenib group and 2.8 months in the placebo group [7]. Importantly, all patients subjected to the treatment with these multi-kinase inhibitors eventually progressed. These clinical findings indicate that resistance to these multi-kinase inhibitors is a major hurdle during HCC treatment. To subvert this gloomy scenario, appropriate models should be established where to test the effectiveness of innovative therapeutic approaches. Thus, in the present investigation, we evaluated the therapeutic efficacy of sorafenib and regorafenib in the AKT/c-MET preclinical HCC model. We discovered that neither sorafenib nor regorafenib slowed HCC progression in vivo. The results are consistent with the clinical observation that only a small percentage of patients with advanced HCC benefit from these regimens, whereas most of the patients either do not or marginally respond to the treatment. It is worth to note that, when dosed at the same concentration, Sorafenib has been found to effectively inhibit cell growth in HCC cell lines and in a xenograft model by blocking the Ras/MEK/MAPK cascade and suppressing angiogenesis [30]. We failed to observe any of these biochemical and cellular effects by sorafenib in vivo using the AKT/c-MET HCC model. In light of the present findings and the scarcity of positive effects by sorafenib and regorafenib on human HCC patients, the present data suggest the AKT/c-MET model as a valid in vivo system to study the mechanisms of resistance to multi-kinase inhibitors in HCC.

AKT/mTOR and Ras/MEK/MAPK signaling pathways are widely upregulated in HCC and could be promising targets in HCC treatment [10,21,28]. Following this hypothesis, first generation mTOR inhibitors, such as everolimus, have been tested in HCC patients. Unfortunately, everolimus failed to show any therapeutic efficacy in clinical trials for advanced HCC [31]. It is important to note that everlimus and other rapalogs that have been tested in clinical trials, are all partial mTORC1 inhibitors. Indeed, they inhibit the activation of the RPS6 protein, but do not affect the 4EBP1/eIF4E axis and the mTORC2 signaling [32,33]. On the other hand, the second generation mTOR inhibitors, such as MLN0128 used in the present study, fully suppress the mTORC1 complex (PRS6 and 4EBP1/eIF4E) as well as mTORC2 [34]. It is also worth to underline that targeting mTOR cascade alone may have limited therapeutic efficacy because tumor cell proliferation could be fully sustained via the compensatory activation of the Ras/MAPK cascade [35,36]. Consequently, concomitant targeting of both AKT/mTOR and Ras/MEK/MAPK signaling pathways may be required for the effective treatment of advanced HCC.

In our previous investigation, we found that AKT/c-MET co-expression promotes activation of the AKT/mTOR and Ras/MAPK pathways in the mouse liver, leading to rapid HCC development [24]. Thus, using this preclinical HCC model, we evaluated the therapeutic potential of the MEK inhibitor and the mTOR inhibitor, either alone or combination, for HCC treatment. We show here that compared to monotherapies, combined treatment with PD901 and MLN0128 induces a more pronounced HCC growth restraint both in vitro and in vivo. Noticeably, both PD901 and MLN0128 single treatments as well as PD901/MLN0128 combination exhibited superior therapeutic efficacy than sorafenib on AKT/c-MET mouse lesions, indicating that the combination of PD901 with MLN0128 might be an effective novel therapy for HCC subsets displaying high expression of c-MET and/or AKT/mTOR and Ras/MEK/MAPK pathways. Nonetheless, due to the poor liver function in most HCC patients, we cannot exclude that the combination of these drugs may be limited by their toxicity. Thus, targeted drug delivery directly into the tumor cells may be necessary. In addition, the combined regimens could be tested via trans-arterial chemoembolization (TACE) to achieve local therapeutic efficacy. Alternatively, siRNA-based therapies targeting members of the MEK1/2 and mTOR pathways might be explored. Overall, while it remains to be determined whether such a combination therapy may be efficacious in the clinical setting, our investigation provides solid preclinical data to support the further investigation of anti-MEK and mTOR based therapies for HCC treatment.

MEK inhibitors may be appropriate to treat cancers with Ras/MEK/ERK pathway activation, which leads to abnormal cell proliferation [21,28]. Furthermore, specific inhibitors or chemotherapeutic drugs that can induce the death of tumor cells may potentiate the anti-cancer efficacy of MEK inhibitors in patients. In our previous study, we revealed that the mTOR inhibitor MLN0128 could suppress intrahepatic cholangiocarcinoma (ICC) development in AKT/YAP mice mainly through the induction of strong apoptosis [29]. The synergistic anti-neoplastic efficacy of combined MEK and mTOR inhibitors has been demonstrated in melanoma, lung, and colorectal cancer, where it resulted in profound tumor cell apoptosis and inhibition of tumor cell proliferation [37,38]. Unfortunately, our study reveals that MLN0128 alone or combined with PD901 treatment fails to induce robust apoptosis in vitro and in vivo, which could explain why the combination therapy was able to induce a stabilization -but not regression- of tumor development in AKT/c-MET mice. As both MEK and mTOR inhibitors promote a decrease in HCC cell proliferation both in vivo and in vitro, the data suggest that these inhibitors could be combined with other small molecules, which may be more potent in inducing apoptosis, for HCC treatment. Some examples include ABT-737 [39], navitoclax [40], and venetoclax [41]. Among them, venetoclax has been approved for the treatment of chronic lymphocytic leukemia with 17p deletions [41]. It would be important to further investigate these apoptosis activators in HCC treatment using preclinical models, and whether they can be combined with MEK or mTOR inhibitors for increased therapeutic efficacy. 

In summary, our findings demonstrate that combined PD901/MLN0128 treatment strongly inhibits tumor growth in AKT/c-MET mice and HCC cell lines. This body of evidence indicates that the combination of anti-MEK and anti-mTOR based therapy could be useful for human HCC treatment.

## 4. Materials and Methods

### 4.1. Reagents

pT3-EF1α, pT3-EF1α-HA-myr-AKT, pT3-EF1α-V5-c-MET, and pCMV/sleeping beauty transposase (pCMV/SB) plasmids were described previously [24,42,43]. An endotoxin-free Maxi Prep Kit (Sigma-Aldrich, St. Louis, MO, USA) was used to purify the plasmids before being injected into mice. Sorafenib, regorafenib, PD0325901 (PD901) and MLN0128 were purchased from LC Laboratories (Woburn, MA, USA).

### 4.2. Hydrodynamic Tail Vein Injection and Mouse Treatment

Female wild-type (WT) FVB/N mice were obtained from Charles River Laboratories (Wilmington, MA, USA). Hydrodynamic injection was performed according to previous study [44]. Briefly, to generate the HCC model, 10μg pT3-EF1α-HA-myr-AKT and 20μg pT3-EF1α-V5-c-Met and 1.2 μg pCMV/SB were injected in FVB/N mice. Sorafenib (30 mg/kg/day), regorafenib (15 mg/kg/day), MLN0128 (0.5 mg/kg/day), PD901 (10 mg/kg/day), MLN0128 + PD901 or vehicle were orally administered by gavage. We started therapy administration 4 weeks post injection for 3 consecutive weeks, and mice were sacrificed 7 weeks after hydrodynamic injection. Total liver weight was recorded and used as the measurement of tumor burden in the study. For sorafenib preparation, 100 mg of the drug were dissolved in 2.5 mL of a stock solution containing 75% ethanol and Cremophor EL (1:1) at 60 °C (40 mg/mL). Subsequently, the drug was vortexed at highest speed and placed back at 60 °C, until sorafenib was completely dissolved. Subsequently, aliquots were frozen and stored at −80 °C. Regorafenib was dissolved in polypropylene glycol 400/polyethylene glycol 400/10% Pluronic F-68 Aqueous solution (42.5:42.5:15) to the concentration of 3 mg/mL. PD901 was dissolved in 0.5% (w/v) hydroxypropyl-methylcellulose (HPMT; Sigma-Aldrich) in water plus 0.2% *v*/*v* Tween 80 to a stock concentration of 3.33 mg/mL. PD901 was orally administered via gavage for 5 days/week. Before gavage, stock solution was diluted with 0.9% NaCl to form a microemulsion. MLN0128 was dissolved in 1-methyl-2-pyrrolidinone (NMP; Sigma-Aldrich) to make a stock solution of 20 mg/mL and the aliquots were stored at −80 °C. It was diluted 1:200 into 15% PVP/H_2_O (PVP: polyvinylpyrrolidone K 30, Sigma-Aldrich; diluted in H_2_O at a 15.8:84.2 *w*/*v* ratio). The diluted solution could be stored at 4 °C for 2–3 weeks in dark. Mice were housed, fed, and monitored in accord with protocols approved by the Committee for Animal Research at the University of California San Francisco (San Francisco, CA, USA), protocol number: AN173073.

### 4.3. Histology and Immunohistochemistry

Mouse liver specimens were fixed overnight in 4% paraformaldehyde (Anatech Ltd, Battle Creek, MI, USA) and embedded in paraffin. Sections were done at 5 μm in thickness. Preneoplastic and neoplastic liver lesions were assessed independently by two board-certified pathologists and liver experts (M.E. and K.U.). Briefly, slides were deparaffinized in xylene, rehydrated through a graded alcohol series, and rinsed in PBS. Depending on the protein target to be revealed, antigen retrieval was achieved by boiling either in 10 mM sodium citrate buffer (pH 6.0) or 1 mM ethylenediaminetetraacetic acid (EDTA; pH 8.5) buffer for 10 min, followed by a 20-min cool down at room temperature. After a blocking step using the 5% goat serum and Avidin-Biotin blocking kit (Vector Laboratories, Burlingame, CA, USA), the slides were incubated with specific primary antibodies (Appendix A) overnight at 4 °C. In order to quench the endogenous peroxidase activity, slides were incubated for 10 min with 3% hydrogen peroxide and, subsequently, the biotin conjugated secondary antibody was applied at a 1:500 dilution for 30 min at room temperature. The immunoreactivity was visualized using the Vectastain Elite ABC kit (Vector Laboratories) and 3, 3′-diaminobenzidine or Vector NovaRed (Vector Laboratories) as the chromogen. Finally, slides were counterstained with hematoxylin. Proliferation index was determined in mouse HCC lesions by counting Ki-67 positive cells on at least 3000 tumor cells per mouse sample. Apoptosis index was determined in mouse HCC lesions by counting TUNEL positive cells on at least 3000 tumor cells per mouse using the TumorTACS^TM^ In Situ Apoptosis Detection Kit (Trevigen, Gaithersburg, MD, USA), following the manufacturer’s protocol. All HCC lesions were carefully analyzed and classified independently by two board-certified pathologists and liver experts (Prof. Matthias Evert and Dr. Kirsten Utpatel).

### 4.4. Western Blot Analysis

Frozen mouse livers tissues and cultured cell samples were homogenized in a lysis buffer consisting of 30 mM Tris (pH 7.5), 150 mM NaCl, 1% NP-40, 0.5% Na deoxycholate, 0.1% SDS, 10% glycerol, and 2mM EDTA] containing the Complete Protease Inhibitor Cocktail (ThermoFisher Scientific, Waltham, MA, USA). For the assessment of protein concentrations, the Bio-Rad Protein Assay Kit (Bio-Rad, Hercules, CA, USA) was employed. Bovine serum albumin was used as standard. For Western blot analysis, aliquots of 40 μg were denatured by boiling in Tris-Glycine SDS Sample Buffer (Bio-Rad), separated by SDS-PAGE, and transferred onto nitrocellulose membranes (Bio-Rad) by electroblotting. Membranes were blocked in Pierce Protein-free Tween 20 Blocking Buffer (ThermoFisher Scientific) for 1 h and then probed with specific antibodies. The complete list of the antibodies used is depicted in Appendix A. Anti-β-Actin (Sigma-Aldrich) and/or GAPDH (EMD Millipore, Burlington, MA) antibody was used as loading control. Each primary antibody was followed by incubation with horseradish peroxidase-secondary antibody (Jackson Immunoresearch Laboratories Inc., West Grove, PA, USA) diluted 1:5000 for 30 min and proteins bands were revealed with the Super Signal West Femto (Pierce Chemical Co., New York, NY, USA).

### 4.5. In Vitro Experiments

SNU475, Huh7, and MHCC97H human HCC cell lines were used for the in vitro studies. The Huh7 cell line was purchased from the JCRB Cell Bank, whereas the SNU475 cell line was purchased from ATCC. MHCC97H cells were a kind gift from Dr. Binbin Liu from Liver Cancer Institute and Zhongshan Hospital of Fudan University, Shanghai, China. Cell lines were maintained as monolayer cultures in Dulbecco’s modified Eagle medium or RPMI 1640 medium supplemented with 10% fetal bovine serum (FBS; Gibco, Grand Island, NY, USA), 100 U/mL penicillin, and 100 g/mL streptomycin (Gibco). 

For IC_50_ determination, cells were seeded in 24-well plates and treated with increasing doses of PD901or MLN0128 in triplicate for 24–48 h. Cells were stained with crystal violet. After washing, stained cells were treated with lysate solution and shaken gently on a rocking shaker for 20–30 min. Diluted lysate solutions were added to 96-well plates and OD was measured at 590 nm with an ELx808 Absorbance Microplate Reader (BioTek, Winooski, VT, USA). All cell line experiments were repeated at least three times in triplicate. 

Cell proliferation was assessed in HCC cell lines at the 24- and 48-hour time points using the BrdU Cell Proliferation Assay Kit (Cell Signaling Technology, Danvers, MA, USA). For the BrdU incorporation assay, control or drug-treated cells were incubated with bromodeoxyuridine (BrdU) for 1.5–3 h and the assay was performed using the FITC BrdU Flow Kit (BD Biosciences, San Jose, CA, USA), following the manufacturer’s instructions. Briefly, the cells were fixed after removing the medium with BrdU. Then, DNase was used to expose incorporated BrdU. Next, the anti-BrdU antibody was added and bound to newly synthesized cellular DNA which is labelled with BrdU. 7-AAD was used for the total DNA staining. The measurement of cell cycle parameters was performed with the Becton Dickinson LSRII Flow Cytometer (BD Biosciences) and data processed using the FlowJo 10 software (FlowJo, LLC, Ashland, OR, USA).

As concerns apoptosis, it was determined in the three HCC cell lines using the Cell Death Detection Elisa plus Kit (Roche Molecular Biochemicals, Indianapolis, IN, USA), following the manufacturer’ instructions. DMSO-, PD901-, and/or MLN0128-treated cells were initially subjected to 24 h serum starvation. Subsequently, apoptotic cell death was assessed at 24 h and 48 h time points. DMSO values were used as the baseline (“1”), and all values were normalized to the baseline reading. All experiments were repeated at least three times in triplicate.

### 4.6. Statistical Analysis

GraphPad Prism version 6.0 (GraphPad Software Inc., La Jolla, CA, USA) was used to evaluate statistical significance. Data are presented as Means ± SD. Comparisons between two groups were performed using U-tests; and multiple groups using ANOVA test. *p* values < 0.05 were considered statistically significant. All in vitro experiments were repeated at least three times in triplicate.

## 5. Conclusions

Our investigation employing HCC cell lines and the AKT/c-MET mouse HCC model suggests that combination of anti-MEK and anti-mTOR inhibitors could be a new therapeutic approach for human HCC treatment.

## Figures and Tables

**Figure 1 cancers-11-00930-f001:**
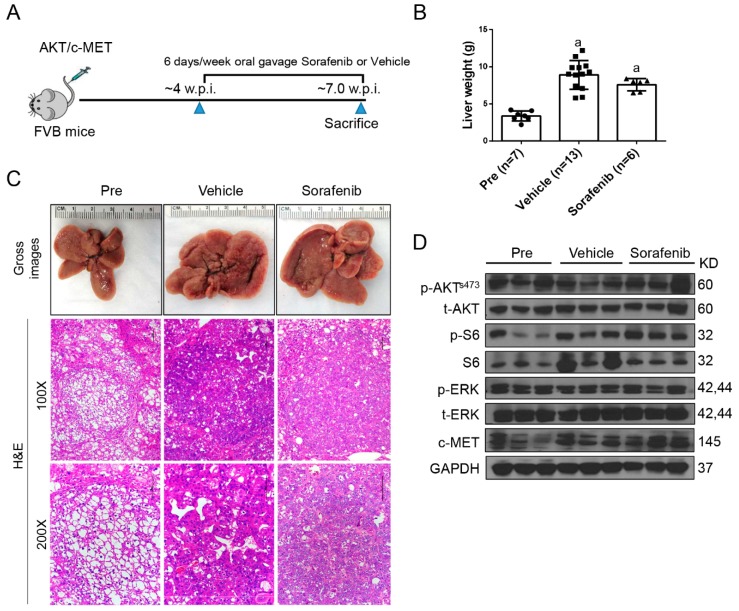
Effect of sorafenib treatment in AKT/c-MET mice. (**A**) Study design. (**B**) Liver weight of pre-treatment, vehicle-, and sorafenib-treated AKT/c-MET mice. (**C**) Gross images and H&E staining of livers from pre-treatment, vehicle-, and sorafenib-treated AKT/c-MET mice. At the pre-treatment stage (Pre), livers of AKT/c-MET mice appear occupied by clear-cell, lipogenic tumors, which are subsequently (vehicle and sorafenib treated groups) substituted by more basophilic lesions. (**D**) Effect of sorafenib administration on the levels of putative target proteins in livers from AKT/c-MET mice. (Magnifications: 100× and 200×, Scale bar: 100 μm). Abbreviations: H&E, hematoxylin and eosin staining; Pre, pre-treatment.

**Figure 2 cancers-11-00930-f002:**
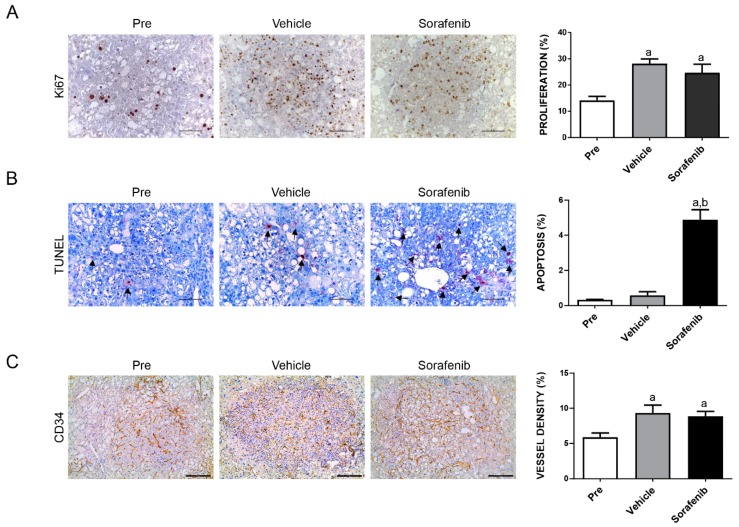
Effects of treatment with sorafenib on the AKT/c-MET mouse lesions, as determined by immunohistochemistry. Ki-67 (**A**) and TUNEL (**B**) staining in livers from AKT/c-MET mice subjected to the various treatments were quantified and represent the percentage of positive cells for proliferation and apoptosis, respectively. At least 3000 tumor cells per sample were evaluated. (**C**) CD34 staining in livers from AKT/c-MET mice subjected to the various treatments. The “vessel density” represents the percentage of CD34 staining area in each field from the sections as assessed by the ImageJ software. Tukey–Kramer test: at least *p* < 0.001. a, vs. Pre-treatment; b, vs. Vehicle. Abbreviations: Pre, pre-treatment.

**Figure 3 cancers-11-00930-f003:**
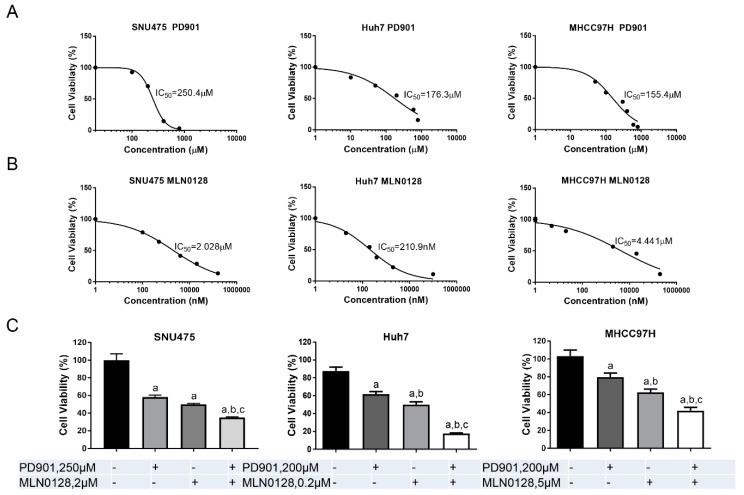
PD901 and MLN0128 inhibit HCC cell growth in vitro. (**A**,**B**) IC_50_ values calculated by quantifying the Crystal violet staining from a panel of HCC cell lines treated for 3 days with the indicated doses of PD901 (**A**) and MLN0128 (**B**). (**C**) Combining PD901 with MLN0128 (around IC_50_ concentration) resulted in a significantly reduced cell viability in HCC cell lines compared with PD901 or MLN0128 single treatment. Abbreviation: Comb, combined PD901/MLN0128 treatment. Tukey–Kramer test: at least *p* < 0.005 a, vs. Control b, vs. PD901; c, vs. MLN0128.

**Figure 4 cancers-11-00930-f004:**
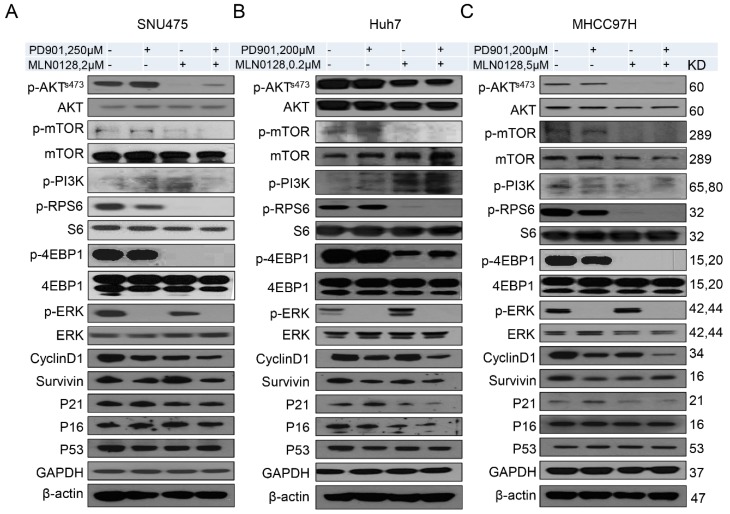
Effect of combined PD901/MLN0128 treatment on the levels of putative targets in HCC cell lines. (**A**–**C**) Representative western blot analysis of AKT/mTOR, Ras/MAPK, and proliferation signaling pathways in SNU475 (**A**), Huh7 (**B**), and MHCC97H (**C**) HCC cell lines.

**Figure 5 cancers-11-00930-f005:**
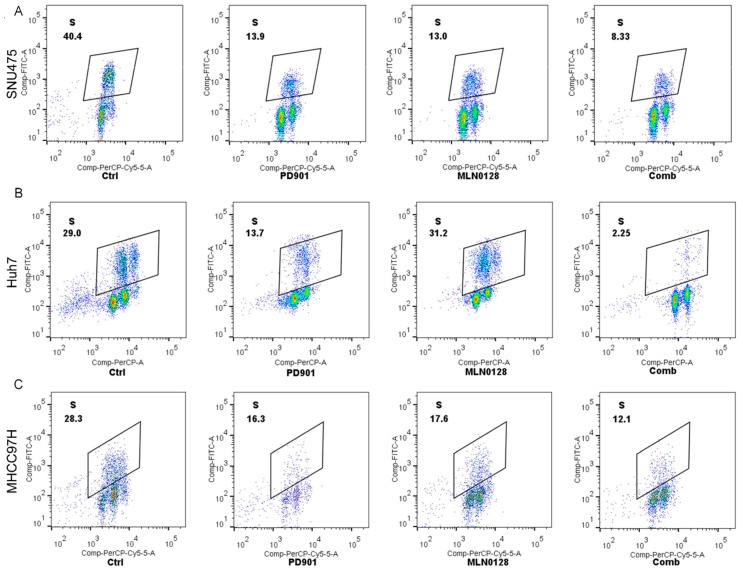
Effect of MLN0128/PD901 combination on cell cycle of HCC cell lines. Enhanced cell cycle arrest in SNU475 (**A**), Huh7 (**B**), and MHCC97H (**C**) cell lines treated with PD901 plus MLN0128 when compared with treatment with PD901 and MLN0128 alone. The percentages of cells in the S phase are shown, together with representative dot plots. Abbreviations: Ctrl, Control; Comb, combined PD901/MLN0128 treatment.

**Figure 6 cancers-11-00930-f006:**
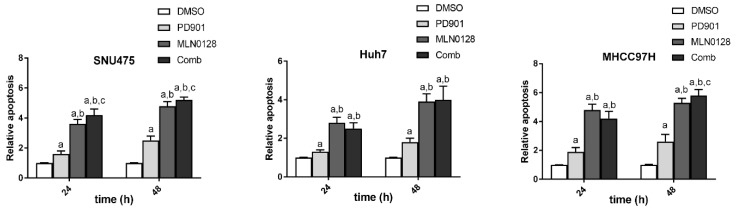
Effect of MLN0128/PD901 combination on apoptosis of HCC cell lines. Enhanced apoptosis in SNU475, Huh7, and MHCC97H cell lines was detected following treatment with either PD901 or MLN0128 when compared to solvent (DMSO). Apoptosis was significantly more pronounced in MLN0128- than PD901-treated cells. No consistent further increase of apoptosis than that observed in PD901- or MLN0128-treated cells was detected when the two drugs were administered in combination. Each bar represents mean ± SD of three independent experiments conducted in triplicate. Tukey-Kramer’s test: *p* at least < 0.005; a, vs. DMSO; b, vs. PD901; c, vs. MLN0128; d, vs. Combination. Abbreviation: Comb, combined PD901/MLN0128 treatment.

**Figure 7 cancers-11-00930-f007:**
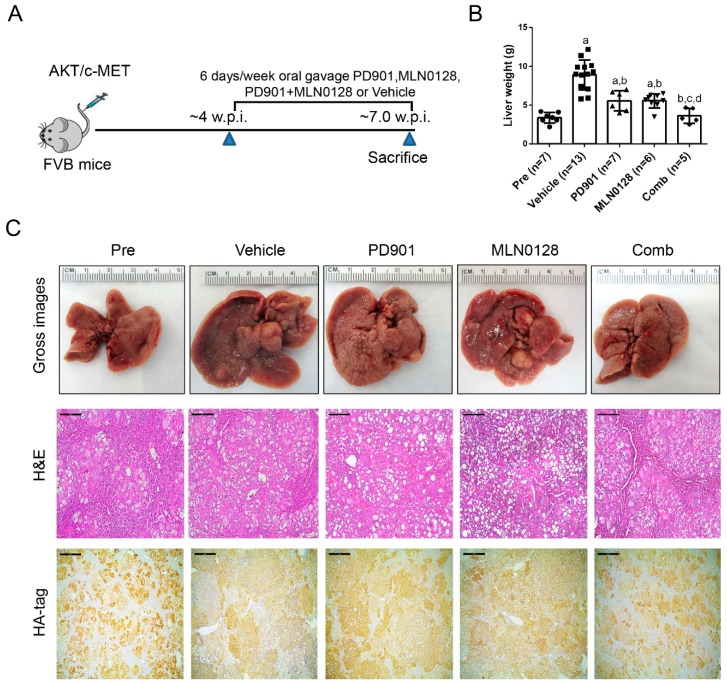
Effects of PD901/MLN0128 combination on hepatocarcinogenesis in AKT/c-MET mice. (**A**) Study design. (**B**) Liver weight of pre-treatment, vehicle-, MLN0128-, PD901-, and PD901/MLN0128-treated AKT/c-MET mice. (**C**) Gross images and H&E and HA-tag staining of livers from pre-treatment, vehicle-, PD901-, MLN0128-, PD901-, PD901/MLN0128-treated AKT/c-MET mice. HA-tag areas indicate the myr-AKT (with a HA-tag) positive cells. Magnification: 100×, Scale bar: 200 μm. Abbreviations: H&E, hematoxylin and eosin staining; Pre, Pre-treatment; Comb, combined PD901/MLN0128 treatment.

**Figure 8 cancers-11-00930-f008:**
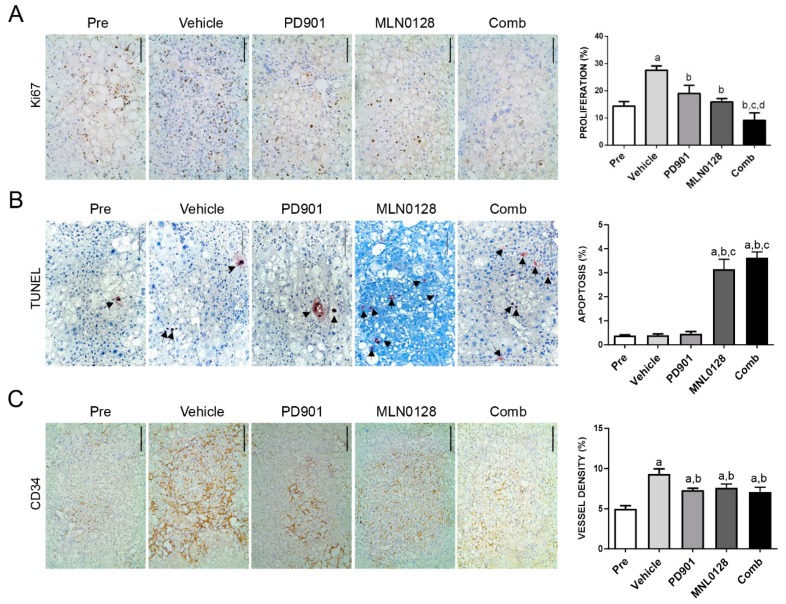
Effect of PD901/MLN0128 combination on the lesions of AKT/c-MET mice, as determined by immunohistochemistry. (**A**) Ki-67 staining in livers from AKT/c-MET mice subjected to the various treatments. Ki-67 positive tumor cells were counted and quantified per 3000 tumor cells. (**B**) TUNEL and CD34 (**C**) staining in livers from AKT/c-MET mice subjected to the various treatments. TUNEL positive tumor cells were counted and quantified per 3000 tumor cells and indicate the apoptosis rate. The “vessel density” represents instead the percentage of CD34 staining area in each field from the sections as assessed by ImageJ software. Tukey–Kramer test: at least *p* < 0.01. a, vs. Pretreatment; b, vs. Vehicle; c, vs. MLN0128; d, vs. PD901. Abbreviations: Pre, Pre-treatment; Comb, combined PD901/MLN0128 treatment.

**Figure 9 cancers-11-00930-f009:**
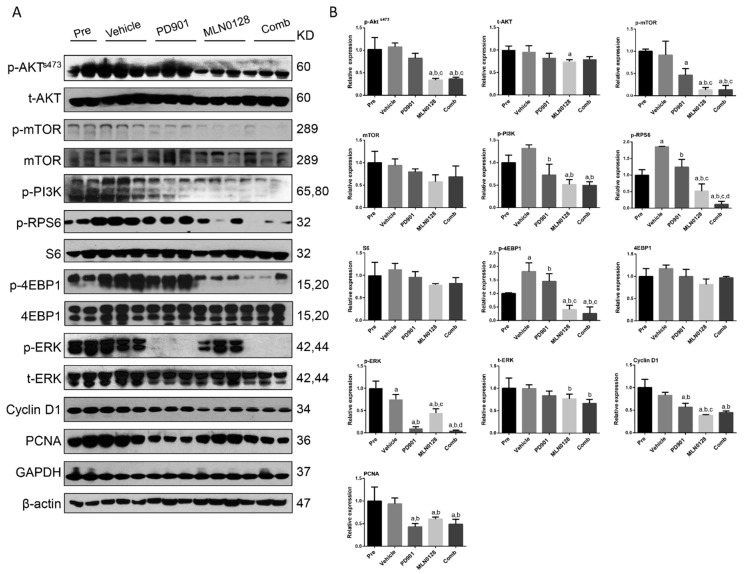
Effect of combined PD901/MLN0128 administration on the levels of putative target proteins in livers from AKT/c-MET mice. Western blot analysis was performed (**A**) and quantified (**B**) to analyze AKT/mTOR, Ras/MAPK, and proliferation pathways, in HCC tissues from pre-treatment, vehicle-, PD901-, MLN0128-, and PD901/MLN0128-treated AKT/c-MET mice. Western blot results were assessed by Image J software. Tukey–Kramer test: at least *p* < 0.01. a, vs Pre; b, vs Vehicle; c, vs PD901; d, vs MLN0128; e, vs Comb. Abbreviations: Pre, Pre-treatment; Comb, combined PD901/MLN0128 treatment.

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
