# Peer review of "Combined Treatment with MEK and mTOR Inhibitors is Effective in In Vitro and In Vivo Models of Hepatocellular Carcinoma"

_cancers, 2019, doi:10.3390/cancers11070930_

Reviewer 1 Report

The background on which the study is based is a mouse model resulting insensitive to sorafenib and intollerant to regorafenib, two of the very rare effective drugs in clinical practise. How to explain this? Do we mean that other pathways must be explored? For sure. But both sorafenib inefficacy and regorafenib intolerance of the model proposed might probably lower the validity of the model itself.

Author Response

Point: The background on which the study is based is a mouse model resulting insensitive to sorafenib and intolerant to regorafenib, two of the very rare effective drugs in clinical practice. How to explain this? Do we mean that other pathways must be explored? For sure. But both sorafenib inefficacy and regorafenib intolerance of the model proposed might probably lower the validity of the model itself.

Response: We would like to clarify that, although Sorafenib and Regorafenib are approved by FDA as first line and second line treatment, respectively, for advanced stage HCC patients, these drugs are not very effective. Indeed, both drugs demonstrate improved survival of ~3 months only for HCC patients, often in the presence of relevant side effects.[1, 2] It is important to notice that the survival data are statistical data from a large cohort of patients. While a significant number of HCC patients do not respond to these drugs or must withdraw due to severe toxicity, a small percentage of patients can survive for longer times (up to one-two years). The ~3-month survival benefit is the average value for all HCC patients enrolled in the clinical trial. In our preclinical HCC model, we show that Sorafenib and Regorafenib are either ineffective or extremely toxic. The observation is highly consistent with the clinical observation that most HCC patients minimally respond or suffer from significant toxicity from these drugs. In addition, Sorafenib and Regorafenib have shown to provide limited benefits in many other tumor types as well, and are recommended for the treatment of very few neoplasms. We would also like to underline that the AKT/c-MET HCC mouse model was generated based on the observation that combined activation of the AKT and c-MET cascades occurs in ~25% of human HCC samples. Our data, therefore, should be highly relevant to the human HCC subsets displaying the activation of AKT and c-MET pathways. We have discussed this important issue in the Discussion section of the revised manuscript (please see page12, Lane 11-13, and page13, 1-3, 6-9).

Reference

1. Llovet JM, Ricci S, Mazzaferro V, Hilgard P, Gane E, Blanc JF, et al. Sorafenib in advanced hepatocellular carcinoma. N Engl J Med. 2008;359: 378-390.

2. Bruix J, Qin S, Merle P, Granito A, Huang YH, Bodoky G, et al. Regorafenib for patients with hepatocellular carcinoma who progressed on sorafenib treatment (RESORCE): a randomised, double-blind, placebo-controlled, phase 3 trial. Lancet. 2017;389: 56-66.

Reviewer 2 Report

Overall, this is a clear, concise, and well-written manuscript.

results

Page 7, lane 8~9, please check it. The patterns of 24 and 48 h in figure 6 are different. please describe in detail. 

Author Response

Point 1: Overall, this is a clear, concise, and well-written manuscript.

Response 1: We deeply thank the Reviewer for the positive comment on our study.

Point 2: Results Page 7, lane 8~9, please check it. The patterns of 24 and 48 h in figure 6 are different. please describe in detail.

Response 2: In accordance with the Reviewer, we have added a detailed description in Results Page 8, Lane 11-16, as follows: “Of note, the combined administration of the two inhibitors did not result in a consistent significant increase of apoptosis when compared with treatment with single agents (Fig. 6). SNU475 cells showed a marginal increased apoptosis in the combination treatment group both at 24h and 48h treatment. As concerns Huh7 cells, there was no significant increased apoptosis in the combination group at 24h and 48h time point. In MHCC97H cells, on the other hand, concomitant PD901 and MLN0128 administration led to a rise in apoptosis rate 48h after treatment (Fig. 6).

Reviewer 3 Report

The present study was performed in order to evaluate the therapeutic efficacy of Sorafenib; Regorafenib; the MEK inhibitor PD901 as well as the pan-mTOR 35 inhibitor MLN0128 in the AKT/c-Met preclinical HCC model. Authors concluded that the combined PD901/MLN0128 treatments  inhibit tumor growth in AKT/c-MET mice and HCC cell lines and the combination of PD901/MLN0128 could be a potentially useful therapeutic modality for human HCC treatment.

Comments

1.      The mail concern is about the present mouse model of HCC. Observing the macro-pictures presented in the paper, I did not find tumors in the mouse livers. How did the authors perform the histopathological analysis? Which place of the liver was chosen for tumor analysis? Furthermore, due to the extremely low quality of figures and microscopic pictures, it is completely impossible to discriminate whether cancer cells are present in the liver of those mice.  

2.      Authors used mouse liver weight as a measurement unit of tumor burden (possibly because there are no visible tumors in the liver). However, the tumor burden refers to the number of tumors or cancers cells, size of the tumor in the body or the amount of cancer in the body! But not the liver weight! Liver weight could be altered by various factors, especially due to treatment with chemicals and greatly vary. From the macro-pictures the liver hypertrophy could be suggested, which could be a result as well as treatment with PD901 and MLN0128, or some other factors.

3.      Another problem is with changing the dose of MLN1028 in vivo experiment. The dose was reduced twice (1mg/kg b.w. to 0.5mg/kg b.w.), therefore, the toxicity, and the hypertrophy of liver cells is likely to be reduced due to the reduction of the dose, but not due to the anticancer effect. As the tumor burden was measures as changes in liver weight, the reduction in liver hypertrophy could be mistakenly considered as a decrease of tumor burden. Therefore, the main conclusion on the effectiveness of combined treatment with PD901 and MLN0128 appeared to be very questionable.

4.      The pictures are of low quality and contain strange horizontal lines. In my version, all pictures demonstrating the results of histological and immunohistochemical evaluations, are completely of focus and must be substituted.

5.      References on the previous study and the method of development of AKT/c-MET mice are very strange. In the reference 36, there is a review on different gene hydrodynamic transfection, but AKT/c-MET was not between them. Furthermore, in the reference 23, MEK inhibitors (U0126 and AZD6244) suppressed HCC cell growth, but there is no information on AKT/c MET hydrodynamic transfection. Is this reference 24? I looked through this paper, and in comparison to the present one, there are good pictures of tumors and histology presented. It could be suggested, that in the present experiment, the hydrodynamic transfection procedures were not repeated carefully, or some mistake occur.

6.      To evaluate the mTOR pathway activation authors measured p-Akt ser473 and p-RP proteins, what is not enough to make conclusion. P-mTOR, p-PI3K are usually also evaluated.

7.      Explanation of Statistical analysis of the data is very short and non-informative. There is no information on how many times in vitro experiments were performed.

8.      In Materials and Methods Western Blotting and Immunohistochemistry sections, the information of antibodies must be included in the text. Hosts of antibodies are also missing. Were they polyclonal or monoclonal raised in rat or rabbit, or mice antibodies? Why the evaluation of caspase 3 positive cells was used for the evaluation of apoptosis? It could be induced not only the caspase pathway. It is better to use methods evaluating DNA fragmentation like ssDNA or the TUNEL.

9.      There is no information from where the cell lines were obtained.

10.  The minor improvement of English of the paper is needed. The abbreviations must be checked and used all through the study. Combo is not good abbreviation for “combination”.

Author Response

Point: The present study was performed in order to evaluate the therapeutic efficacy of Sorafenib; Regorafenib; the MEK inhibitor PD901 as well as the pan-mTOR inhibitor MLN0128 in the AKT/c-Met preclinical HCC model. Authors concluded that the combined PD901/MLN0128 treatment inhibit tumor growth in AKT/c-MET mice and HCC cell lines and the combination of PD901/MLN0128 could be a potentially useful therapeutic modality for human HCC treatment.

Response: We sincerely thank the Reviewer for the appreciation of our study.

Point 1: The mail concern is about the present mouse model of HCC. Observing the macro-pictures presented in the paper, I did not find tumors in the mouse livers. How did the authors perform the histopathological analysis? Which place of the liver was chosen for tumor analysis? Furthermore, due to the extremely low quality of figures and microscopic pictures, it is completely impossible to discriminate whether cancer cells are present in the liver of those mice.  

Response 1: We deeply apologize for the low quality of the microscopic pictures, which was partly the consequence of embedding the original pictures (TIFF files) in the Word file. Such inadequate quality of the pictures significantly hampered the possibility of the Reviewer to identify the HCC lesions. In agreement with the Reviewer, we replaced the former pictures in Figure 1 with new ones of better quality and in two distinct magnifications. In addition, we would like to emphasize that in the Pre-treatment group (4 weeks post hydrodynamic gene delivery) the tumor lesions are still relatively small and consist mainly of lipid-rich tumor cells. This time point was selected as a starting point of the treatment because liver neoplastic lesions are already present but do not jeopardize the survival of AKT/c-MET mice. Finally, we would like to underline that the liver lesions developing in this mouse model were carefully analyzed and classified independently by two board-certified pathologists and liver experts (Prof. Matthias Evert and Dr. Kirsten Utpatel). We made this point clear in the revised Materials and Method section (please see page 15, Lane 6-7).

Point 2: Authors used mouse liver weight as a measurement unit of tumor burden (possibly because there are no visible tumors in the liver). However, the tumor burden refers to the number of tumors or cancers cells, size of the tumor in the body or the amount of cancer in the body! But not the liver weight! Liver weight could be altered by various factors, especially due to treatment with chemicals and greatly vary. From the macro-pictures the liver hypertrophy could be suggested, which could be a result as well as treatment with PD901 and MLN0128, or some other factors.

Response 2: We appreciate this comment. AKT/c-MET mice develop HCC lesions that tend rapidly to merge into a unique tumor mass in the absence of a capsule separating the original tumor nodules. This tumor mass rapidly occupies over 90% of the liver parenchyma. Consequently, it is extremely difficult to determine the tumor size and/or tumor number in this model. Thus, the liver weight is used as tumor burden in this mouse model. Liver weight is commonly used as a measurement of liver tumor burden for mouse HCC lesions generated via hydrodynamic transfection.[1,2] As concerns toxicity, we monitored the mouse status and measured the mouse weight along the course of the treatment. Mice appeared to be active and no loss of body weight was detected. Importantly, despite the small percentage of non-tumorous liver in these mice, the non-tumorous hepatocytes appeared to be normal, supporting the limited toxicity of the combination therapy. We have made this point clear in the revised Result section (please see page3, Lane 23-28).

Point 3: Another problem is with changing the dose of MLN1028 in vivo experiment. The dose was reduced twice (1mg/kg b.w. to 0.5mg/kg b.w.), therefore, the toxicity, and the hypertrophy of liver cells is likely to be reduced due to the reduction of the dose, but not due to the anticancer effect. As the tumor burden was measures as changes in liver weight, the reduction in liver hypertrophy could be mistakenly considered as a decrease of tumor burden. Therefore, the main conclusion on the effectiveness of combined treatment with PD901 and MLN0128 appeared to be very questionable.

Response 3: We agree that the dose of MLN0128 was halved from 1mg/kg/day to 0.5mg/kg/day. In our study, the combined MLN0128 and PD901 treatment at the dose indicated was well tolerated, as there was no mouse body weight loss and the remaining non-tumorous hepatocytes appear to be normal. We also would like to emphasize that the anti-tumor activity of combined MLN0128 and PD901 treatment was not based on the mouse liver weight alone. All HCC cells stained positive for HA tag (the ectopically expressed myr-AKT had HA tag on the plasmid), and HA tag positive HCC lesions grew slower in MLN0128 and PD901 treated mice. Furthermore, Ki-67 staining demonstrated the significant decreased tumor cell proliferation rates following the various treatments. Altogether, these data support our conclusion that combined treatment of PD901 and MLN0128 lead to decreased tumor growth, but not general toxicity to normal hepatocytes. We have made this point clear in the Result section (please see page9, Lane 26-39 and page10, Lane 13-18).

Point 4: The pictures are of low quality and contain strange horizontal lines. In my version, all pictures demonstrating the results of histological and immunohistochemical evaluations, are completely out of focus and must be substituted.

Response 4: We deeply apologize for the low quality of the microscopic pictures, which was partly the consequence of embedding the original pictures (TIFF files) in the Word file. Indeed, the “strange horizontal lines” are not present in the original TIFF files. In agreement with the Reviewer, we replaced the former pictures in Figure 1 with new ones of better quality and in two distinct magnifications.

Point 5: References on the previous study and the method of development of AKT/c-MET mice are very strange. In the reference 36, there is a review on different gene hydrodynamic transfection, but AKT/c-MET was not between them. Furthermore, in the reference 23, MEK inhibitors (U0126 and AZD6244) suppressed HCC cell growth, but there is no information on AKT/c MET hydrodynamic transfection. Is this reference 24? I looked through this paper, and in comparison to the present one, there are good pictures of tumors and histology presented. It could be suggested, that in the present experiment, the hydrodynamic transfection procedures were not repeated carefully, or some mistake occur.

Response 5: We would like to thank the Reviewer for this comment. Reference 36 is a method paper for the generation of novel mouse models using the hydrodynamic transfection technique for liver cancer research. We try to build different new liver cancer models by injecting different combinations of oncogenic plasmids into mice tail vein according to the method introduced in this paper. The AKT/c-MET is one of the “pure” (developing exclusively HCC lesions) HCC mouse models that we established successfully. We sincerely apologize for the mistake of reference 23; consequently, we have changed the reference 23 to reference 24 in page2, line 47 and page5, line 12, in the revised manuscript.

Point 6: To evaluate the mTOR pathway activation authors measured p-Akt ser473 and p-RP proteins, what is not enough to make conclusion. P-mTOR and p-PI3K are usually also evaluated.

Response 6: We would like to thank the Reviewer for this excellent suggestion and added p-mTOR and p-PI3K blots in Figure 4 and Figure 9. We have made this point clear in the revised Result section (please see page6, Lane 8-11 and page11, Lane 18-19).

Point 7: Explanation of Statistical analysis of the data is very short and non-informative. There is no information on how many times in vitro experiments were performed.

Response 7: We thank the Reviewer for this suggestion. We have added the explanation of Statistical analysis and in vitro experiments information in the Materials and Methods on page16 Lane 6-8.

Point 8: In Materials and Methods Western Blotting and Immunohistochemistry sections, the information of antibodies must be included in the text. Hosts of antibodies are also missing. Were they polyclonal or monoclonal raised in rat or rabbit, or mice antibodies? Why the evaluation of caspase 3 positive cells was used for the evaluation of apoptosis? It could be induced not only the caspase pathway. It is better to use methods evaluating DNA fragmentation like ssDNA or the TUNEL.

Response 8: We thank the Reviewer for these insightful suggestions. Accordingly, we have added the antibodies missing information in Supplementary Table 1. Furthermore, in agreement with the Reviewer, we performed TUNEL staining for the evaluation of apoptosis and showed the results in Figure 2 and 8 of the revised manuscript, replacing the cleaved caspase 3 data. As expected, the results obtained with TUNEL staining were similar to those previously obtained with cleaved caspase 3 immunohistochemistry.

Point 9: There is no information from where the cell lines were obtained.

Response 9: We thank the Reviewer for this suggestion and added the related information on page15 Lane 26-29: “The Huh7 cell line was purchased from JCRB Cell Bank. The SNU475 cell line was purchased from ATCC. MHCC97H cells were a kind gift from Dr. Binbin Liu from Liver Cancer Institute and Zhongshan Hospital of Fudan University, Shanghai, China.”

Point 10: The minor improvement of English of the paper is needed. The abbreviations must be checked and used all through the study. Combo is not good abbreviation for “combination”

Response 10: We thank the Reviewer for this suggestion. We improved the text and checked the abbreviations in the manuscript. In addition, in agreement with the Reviewer, we changed the abbreviation of “combination” from “Combo” to “Comb” in Figure 5, 7 and 8 and the corresponding figure legends.

References

1. Qiu Z, Zhang C, Zhou J, Hu J, Sheng L, Li X, et al. Celecoxib alleviates AKT/c-Met-triggered rapid hepatocarcinogenesis by suppressing a novel COX-2/AKT/FASN cascade. Mol Carcinog. 2019;58: 31-41.

2. Huntzicker EG, Hotzel K, Choy L, Che L, Ross J, Pau G, et al. Differential effects of targeting Notch receptors in a mouse model of liver cancer. Hepatology. 2015;61: 942-952.

Reviewer 4 Report

In the manuscript from Liu et al., the authors provide a substantial amount of data with high quality from in vitro and in vivo experiments performed in HCC. They show convincingly that a combination of  MEK and mTOR nhibitors causes growth inhibition and apoptosis of HCC cells.

I find the data to be of high quality and to be of interest for the cancer research field. 

Author Response

In the manuscript from Liu et al., the authors provide a substantial amount of data with high quality from in vitro and in vivo experiments performed in HCC. They show convincingly that a combination of MEK and mTOR inhibitors causes growth inhibition and apoptosis of HCC cells.

Point: I find the data to be of high quality and to be of interest for the cancer research field.

Response: We sincerely thank the Reviewer for the appreciation of our study.

Round  2

Reviewer 1 Report

The paper is interesting but the core of it presents some main debatable issues.

In fact, at the present knowledge, m-Tor inhibitors have proven to be futile as HCC  treatment. Moreover, as a proof of principle, the traslation of the efficacy of two different class drugs in mice, to the human model, is far from obvious.

Furthermore, although the nice scientific soundness of the paper, it is hard to accept, due to the futility shown by m-Tor inhibitors as HCC treatment, that the best research strategy for HCC therapy would be to associate m-Tor inhibitors with other class drugs.

Author Response

Point: The paper is interesting but the core of it presents some main debatable issues.

In fact, at the present knowledge, m-Tor inhibitors have proven to be futile as HCC treatment. Moreover, as a proof of principle, the translation of the efficacy of two different class drugs in mice, to the human model, is far from obvious.

Furthermore, although the nice scientific soundness of the paper, it is hard to accept, due to the futility shown by m-Tor inhibitors as HCC treatment that the best research strategy for HCC therapy would be to associate m-Tor inhibitors with other class drugs.

Response: We thank the Reviewer for the thoughtful comment. Nonetheless, we regretfully disagree at least in part with the Reviewer’s conclusions. Indeed, we would like to clarify that the mTOR inhibitors used in clinical trials for hepatocellular carcinoma so far were mTORC1 inhibitors (rapamycin and rapalogs), which are only able to partially suppress the mTORC1 pathway. Indeed, these drug effectively block the activity of RPS6 protein, but do not affect the second branch of the mTORC1 pathway, namely the 4EBP1/eIF4e axis [1, 2], but rather contribute to an escaping mechanism by the PI3K-mTORC2-AKT pathway [3]. In contrast, the new mTOR inhibitors, such as MLN0128 used in the current study, fully suppress the mTORC1 complex (RPS6 and 4EBP1/eiF4E) as well as mTORC2 [4, 5]. Currently, these drugs are in clinical trials for HCC treatment, and there is no evidence that these pan-mTOR inhibitors do not work [2]. In addition, it is highly likely that the previous trials using rapalogs failed because the drug was administered to people without prior selection of patient populations based on molecular classification or predictive biomarkers, such as whether their HCC were mTOR positive or negative [2]. Furthermore, the mTOR novel inhibitors have never been tested so far in association with MEK inhibitors in human HCC. Therefore, altogether our data provide solid evidence indicating that this combination treatment is highly detrimental for the growth of HCC cells in vitro and in vivo, supporting further investigation on drugs targeting these pathways for human HCC treatment. We have added the discussion of this issue to the Discussion Section of the revised manuscript (please refer to page13, lane 6-17).

References

1. Martelli AM, Buontempo F, McCubrey JA. Drug discovery targeting the mTOR pathway. Clin Sci (Lond). 2018;132: 543-568.

2. Yu XN, Chen H, Liu TT, Wu J, Zhu JM, Shen XZ. Targeting the mTOR regulatory network in hepatocellular carcinoma: Are we making headway? Biochim Biophys Acta Rev Cancer. 2019;1871: 379-391.

3. Matter MS, Decaens T, Andersen JB, Thorgeirsson SS. Targeting the mTOR pathway in hepatocellular carcinoma: current state and future trends. J Hepatol. 2014;60: 855-865.

4. Polivka J, Jr., Janku F. Molecular targets for cancer therapy in the PI3K/AKT/mTOR pathway. Pharmacol Ther. 2014;142: 164-175.

5. Zhang YJ, Duan Y, Zheng XF. Targeting the mTOR kinase domain: the second generation of mTOR inhibitors. Drug Discov Today. 2011;16: 325-331.

Reviewer 3 Report

Authors have performed additional experiments and corrected the manuscript accordingly to the suggestions. Therefore,  it can be now accepted for publication. Microscopical pictures are much better now. Results and Materials and Methods are now presented in details.

Author Response

Point: Authors have performed additional experiments and corrected the manuscript accordingly to the suggestions. Therefore, it can be now accepted for publication. Microscopical pictures are much better now. Results and Materials and Methods are now presented in detail.

Response: We sincerely thank the Reviewer for the appreciation of our study and the modifications employed.